# Multibranch Wavelet-Based Network for Image Demoiréing

**DOI:** 10.3390/s24092762

**Published:** 2024-04-26

**Authors:** Chia-Hung Yeh, Chen Lo, Cheng-Han He

**Affiliations:** 1Department of Electrical Engineering, National Taiwan Normal University, Taipei 10610, Taiwan; 81075005h@ntnu.edu.tw (C.L.);; 2Department of Electrical Engineering, National Sun Yat-sen University, Kaohsiung 80424, Taiwan

**Keywords:** moiré pattern, wavelet transform, deep learning, image restoration

## Abstract

Moiré patterns caused by aliasing between the camera’s sensor and the monitor can severely degrade image quality. Image demoiréing is a multi-task image restoration method that includes texture and color restoration. This paper proposes a new multibranch wavelet-based image demoiréing network (MBWDN) for moiré pattern removal. Moiré images are separated into sub-band images using wavelet decomposition, and demoiréing can be achieved using the different learning strategies of two networks: moiré removal network (MRN) and detail-enhanced moiré removal network (DMRN). MRN removes moiré patterns from low-frequency images while preserving the structure of smooth areas. DMRN simultaneously removes high-frequency moiré patterns and enhances fine details in images. Wavelet decomposition is used to replace traditional upsampling, and max pooling effectively increases the receptive field of the network without losing the spatial information. Through decomposing the moiré image into different levels using wavelet transform, the feature learning results of each branch can be fully preserved and fed into the next branch; therefore, possible distortions in the recovered image are avoided. Thanks to the separation of high- and low-frequency images during feature training, the proposed two networks achieve impressive moiré removal effects. Based on extensive experiments conducted using public datasets, the proposed method shows good demoiréing validity both quantitatively and qualitatively when compared with the state-of-the-art approaches.

## 1. Introduction

Moiré pattern is a type of visual artifact that is noted when interference occurs between overlapping patterns, such as the grid of display elements and camera sensors, greatly reducing the overall perceptual quality of the image. With the development of digital imaging technology and the widespread use of digital cameras and displays, images with moiré patterns have become increasingly common. The appearance of moiré patterns is quite diverse and complex, forming different stripes, curves, or ripples in space, and the patterns are also accompanied by color changes superimposed on the image. Due to its complexity, moiré pattern removal remains a challenging problem that is attracting increasing attention from the research community. Unlike other image restoration tasks, such as denoising, dehazing, and super-resolution, is difficult to synthesize the moiré effect it into a formation model due to its complex nature. Moiré patterns exhibit irregularities in shape and color over a wide range of frequencies. Figure 1 shows moiré textures of different scales, frequencies, and colors.

Moiré patterns are caused by frequency aliasing. Anti-aliasing optical filters (low-pass filters) can solve this problem; however, it is easy to over-smooth the image, resulting in a blurred image [1]. Several studies have used signal processing techniques to remove moiré patterns [2]. Wei et al. used local median filters and Gaussian notch filters to remove moiré patterns [3]; however, the process is a semi-automatic and requires manual parameter adjustment. Liu et al. proposed a low-rank and sparse matrix decomposition model for image demoiréing; however, the complex pattern still cannot be effectively eliminated. Yang et al. [4] used a low-rank sparse matrix decomposition model to eliminate moiré patterns in the green channel; guided filtering is applied to the extracted green channel texture layer to remove moiré patterns in the red and blue channels. However, it is unable to process screen captured images. Yang et al. [5] proposed a demoiréing method by applying Layer Decomposition Polyphase Component (LDPC) to screen capture images, which results in high computational complexity and excessive smoothing of image details.

In order to achieve better moiré removal performance, a learning framework for single image moiré removal has been presented in [6,7,8,9,10,11,12,13,14,15,16] based on the deep study of moiré patterns from a large dataset. Relying on the rapid development of deep learning with great success in numerous perceptual tasks, especially in image restoration, several deep learning-based single image moiré removal frameworks have been presented. They have shown that moiré-relevant features are automatically learned using a deep convolutional neural network to achieve better results in image moiré removal.

This paper proposes to remove moiré patterns in the frequency domain by initially decomposing the moiré image into distinct frequency bands using discrete wavelet transform (DWT). Moiré patterns come in many forms and exist in both high- and low-frequency images, so different methods are required to effectively remove moiré patterns. Sub-band images, the high- and low-frequency images obtained through wavelet decomposition, are then fed to different branches, which use networks for end-to-end learning to effectively remove different levels of moiré patterns. The low-frequency sub-images at each decomposition level are processed using the moiré removal network (MRN) to eliminate moiré effects in the low-frequency band, while preserving background details. The high-frequency sub-images are treated with detail-enhanced moiré removal network (DMRN) to remove moiré patterns, effectively retaining high-frequency details. The inherent reversibility of the wavelet transformation can prevent the loss of image information during the upsampling and downsampling in learning procedures, thus ensuring a sufficient receptive field. Therefore, the proposed method is expected to better learn moiré patterns than current deep learning-based image demoiréing approaches. Furthermore, with the integration of a multibranch process, better demoiréing results will be produced. Experimental results show that our method outperforms the state-of-the-art moiré removal methods in terms of quantitative and qualitative comparisons.

The remainder of the paper is organized as follows. In Section 2, we provide an overview and discuss work related to moiré pattern removal. Section 3 explains the proposed framework of the moiré removal network. In Section 4, we present the experimental results. Finally, concluding remarks are made in Section 5.

## 2. Background Review

This section provides an overview of the most relevant work on image demoiréing. First, image restoration using deep learning techniques is briefly reviewed. Then, we review some deep learning-based demoiréing methods that inspired the design of our proposed deep model. In addition, frequency-based demoiréing methods are also studied in this section.

### 2.1. Image Restoration

Traditional image restoration methods are usually based on manual feature extraction and models, but these methods may be limited when dealing with complex image problems. Recently, deep learning methods have received a great deal of attention in almost all fields of image processing, especially for image restoration tasks such as image denoising, deblurring, dehazing, and super-resolution. In deep learning, some famous structures such as U-Net [17], recurrent neural networks [18], multi-stage network [19], MIMO-UNet [20], and encoder-decoder transformer [21] have performed well in image restoration tasks. These methods are often able to learn effective feature representations from large amounts of data, enabling them to better handle image restoration problems [22,23,24,25,26,27,28,29]. In addition, mechanisms such as attention modules, residual learning, and multiscale representation are introduced into the deep learning architecture to further improve the efficiency of the restoration model. Through deep learning, the model can better understand and learn the internal structure of the image, thereby achieving more accurate and efficient image restoration.

### 2.2. Moiré Pattern Removal Using Deep Learning

Several image demoiréing methods have been proposed to treat image demoiréing as an image restoration problem, capable of solving a wider range of moiré pattern types. Sun et al. [6] proposed a multiscale CNN (DMCNN) to remove multi-frequency moiré patterns and established a large-scale benchmark dataset based on ImageNet [30] called the TIP dataset. Liu et al. [7] constructed a convolutional neural network with coarse- and fine-scale options. In the coarse-scale network, the input image is first downsampled and passed through stacked residual blocks to remove moiré patterns. The fine network upsamples the processed low-resolution image back to its original resolution. He et al. [8] proposed MopNet, which was specifically designed for moiré patterns, including multiscale feature aggregation for resolving complex frequencies, edge detection to explore the degree of imbalance between color channels, and a perceptual classifier designed to represent the appearance of moiré stripe diversity. Yu et al. [9] introduced a method to demonstrate moiré patterns in 4k ultra high-definition images (UHDM). Yang et al. [10] proposed a parallel multiscale architecture with residual blocks. To efficiently exploit the relationship between different feature maps, this architecture employs an information exchange module (IEM) and a final feature fusion layer (FFF) to continuously exchange information throughout the network. To eliminate multiscale moiré patterns, they built a semantically aligned scale-aware module. Yang et al. [11] proposed a new strategy to flexibly model moiré patterns, enabling the construction of well-aligned training triplets. These triplets were utilized by the translation network MoireDet for moiré pattern detection. Niu et al. [12] proposed a progressive moiré removal and texture complementation network. Their method used a multi-stream framework to gradually remove moiré patterns from low- to high-resolution images. They proposed a progressive texture complementation block to fuse information from different streams, improving moiré removal and detail complementation. Nguyen et al. [13] proposed a multiscale guided screenshot demoiréing algorithm to understand moiré frequency correlations. Initially, they extracted multiscale features from the input image. Then, they introduced FTB and MGRB to capture and remove moiré artifacts at each scale by modulating features. Although some published algorithms perform well, the problem remains largely unresolved due to significant changes in the frequency, shape, and color of moiré patterns.

### 2.3. Frequency Domain Learning

Frequency analysis has always been a powerful tool in image processing that can effectively use frequency domain information to significantly improve the performance of image restoration. Transform operations are often used as image pre- or post-processing, and downsampling or upsampling processes are also used to design deep networks. Some methods embed frequency information into the network structure to exploit its availability [31,32,33,34,35,36,37]. Liu et al. [36] proposed a framework based on Generative Adversarial Networks (GAN) to embed facial attribute vectors in the generator and discriminator. This method guides the model to generate facial images with corresponding input attributes. To enhance aging details, they employed wavelet packet transform to effectively extract features at multiple scales in frequency space. Li et al. [37] proposed a method to transform DWT and IDWT into a general network layer, replacing the traditional upsampling and downsampling operations. The wavelet integrated convolutional network (WaveCNet) was introduced for image classification and achieves higher image classification accuracy and better noise suppression. Several works focus on moiré pattern removal using frequency domain processes. Zheng et al. [14] proposed a learnable bandpass filter (MBCNN) to first learn the frequency to eliminate moiré patterns and designed a tone mapping strategy for color restoration. Although removing moiré patterns using frequency domain processing has achieved good results in deep learning networks, over-smoothing high frequency details and distinguishing low frequency patterns from background colors in complex background images remain a challenge. Liu et al. [15] proposed a two-branch wavelet network with spatial attention mechanism for eliminating moiré patterns, and they utilized dense convolution and dilated convolution to achieve large-scale sensing fields. Sun et al. [16] replaced conventional downsampling and upsampling with DWT and IDWT and introduced an efficient attention fusion module to enable the network to dynamically prioritize feature information, with a specific focus on moiré details. Wavelet transform has several advantages, including time–frequency localization, multi-resolution analysis, and noise reduction. For time–frequency localization, wavelets can be used to identify specific features in a signal, such as transient events or harmonics, that may not be visible with other time–frequency localization methods. Moiré patterns are characterized by higher harmonics and transient events, and wavelet transform is useful for identifying these specific features. The wavelet transform provides a multi-resolution representation of the signal, which means that the moiré signal can be analyzed at different resolution levels so that we can use different strategies to process it. Moreover, wavelet transform can separate signals and noise in different frequency bands, making it easier to suppress noise in moiré patterns. The proposed multibranch network combined with wavelet transform is able to extract more meaningful and sparse moiré pattern representations.

## 3. Proposed Method

In this section, we first explain the embedding method used for the wavelet transform and its inverse. We then introduce two key components: moiré removal network (MRN) and detail-enhanced moiré removal network (DMRN). These two networks play a crucial role in moiré removal and image detail enhancement. Finally, we define the optimization model tailored to our specific loss function. This section aims to provide a concise yet informative overview of the key techniques and components used in our image restoration approach.

### 3.1. Wavelet-Based Image Decomposition

Wavelet transform decomposes an image into multiple sub-images, which are also wavelet coefficients and can be used as features for different tasks. The traditional DWT is translation variant; therefore, its effectiveness may be negatively affected. We select translation-invariant wavelet transform to ensure the robustness of sub-images. Given an image X, X is decomposed into four sub-band images, called $X_{LL},\;X_{LH},\;X_{HL},$ and $X_{HH}$ using the 2D DWT [38], which consists of four filters, namely $f_{LL},\;f_{LH},\;\;f_{HL},\;{\mathrm{and}\;f}_{HH}$. During the transformation, these four filters have fixed parameters and use a convolution step size of 2. Taking the Haar wavelet as an example, these four filters are defined as follows:(1)$$f_{LL}=\left[\begin{array}{cc} 1 & 1\\ 1 & 1 \end{array}\right],\;f_{LH}=\left[\begin{array}{cc} -1 & -1\\ 1 & 1 \end{array}\right],\;f_{HL}=\left[\begin{array}{cc} -1 & 1\\ -1 & 1 \end{array}\right],\;f_{HH}=\left[\begin{array}{cc} 1 & -1\\ -1 & 1 \end{array}\right].$$

DWT reduces the resolution of the input images, making it easier to extract features of image structure and texture. If the size of the image is n×n×3, it becomes (n/2)×(n/2)×12 after DWT, and downsampling is also performed to make the feature channel four times the original one. We decompose the moiré image M using the Haar DWT to obtain wavelet bands at three different levels. The sub-images of each level can be denoted as follows:(2)$$DWT\left(M\right)=\left[I_{LL}^{1},\;I_{LH}^{1},\;I_{HL}^{1},\;I_{HH}^{1}\right],\;DWT\left(I_{LL}^{n}\right)=\left[I_{LL}^{n+1},\;\;I_{LH}^{n+1},\;I_{HL}^{n+1},\;I_{HH}^{n+1}\right],\;\;n\in (1,2,3)$$
where DWT is the wavelet decomposition operation, and I is the sub-image obtained through DWT decomposition. $I_{LL}^{n},\;\;I_{LH}^{n},\;I_{HL}^{n},$ and $I_{HH}^{n}$ are the sub-images bands LL, LH, HL, and HH at level n, respectively. Specifically, LL has the overall context of the image, while LH, HL, and HH encapsulate the detailed information of the image.

### 3.2. Multibranch Network Modules

As shown in Figure 2, the moiré pattern image is decomposed using DWT into three levels of sub-images. At each level, low-frequency and high-frequency sub-images are processed by the moiré removal network (MRN) and a detail-enhanced moiré removal network (DMRN), respectively, in a process called one branch learning. To enable the network to learn more complex mappings, the high-frequency and low-frequency sub-images processed using DMRN and MRN of the current branch learning will be combined through IDWT, and the results will be sent to the next branch for learning. Here, the number of the branch is equal to that of the DWT decomposition level. For example, at the level n, both high-frequency and low-frequency sub-images are processed using the pipeline process, branch n. The output of branch n includes the low-frequency sub-image of level n − 1, which is processed by branch n − 1 together with the high-frequency sub-image of level n − 1. This looping process continues on all branches until the output size matches the original input, thereby enhancing the network’s ability to capture complex features and patterns. The multibranch learning process can be represented as follows:(3)$$R_{LL}^{n}=MRN(I_{LL}^{n}),\;\left[R_{LH}^{n},\;R_{HL}^{n},\;R_{HH}^{n}\right]=DMRN\left[\;I_{LH}^{n},\;I_{HL}^{n},\;I_{HH}^{n}\right],\;\;{Out}_{n}=IDWT(R_{LL}^{n},\;R_{LH}^{n},\;R_{HL}^{n},\;R_{HH}^{n}),\;\;n\in (1,2,3)$$
where $R_{LL}^{n}$ and $\left(R_{LH}^{n},\;R_{HL}^{n},\;R_{HH}^{n}\right)$ are the demoiréd low-frequency and high-frequency images obtained by the nth branch learning, and ${Out}_{n}$ is the output image of the nth branch learning through IDWT.

### 3.3. Moiré Removal Network

The objective of MRN is to remove moiré patterns from the low-frequency image decomposed by DWT while preserving the structure of smooth areas. We extract features through the hierarchical structure, which has always been a common method to solve problems at different scales. The hierarchical structure, which includes two convolutional layers with 3 × 3 filters and 32, 64, and 128 output channels for branches 1, 2, and 3 learning, aims to extract multiscale features within the same level and allow them to interact and dynamically fuse, significantly improving the model’s ability to handle moiré patterns at various scales. Then, eight residual blocks (ResBlocks) [39] are used to prevent the problem of gradient vanish and gradient explosion in deep networks. Squeeze-and-excitation block (SEBlock) [40], the attention module we used, consists of squeeze, excitation, and scale, which is simple yet effective in helping the model learn to focus more on important features. Finally, after passing through eight ResBlocks again, the channel size gradually decreases, finally returning to three channels.

### 3.4. Detail-Enhanced Moiré Removal Network

The objective of DMRN is to remove moiré patterns from the high-frequency image decomposed by DWT while enhancing the fine details in the image. DenseNet [41] has demonstrated excellent performance in various image restoration tasks; specifically, it effectively exploits hierarchical features by densely connecting multiple residual dense blocks (RDBs). We leverage four RDBs to efficiently train the model simultaneously to learn high-frequency details and eliminate moiré patterns. These RDBs embody residual connections and dense feature extraction, which help capture complex patterns and enhance the model’s ability to process high-frequency information. The combination of multiple RDBs further enables the network to effectively learn and represent complex visual features, ultimately helping to eliminate high-frequency moiré patterns during training as shown in Figure 3. To guide the network to efficiently discern high-frequency details from high-frequency moiré patterns, our network focuses on directly connecting inputs and outputs through skip connections, learning only the residual information between input and output.

### 3.5. Loss Function

To learn the proposed network, the problem is formulated as a mapping function between a moiré image component and its corresponding moiré-free component. To solve the problem, we intend to learn a mapping function subject to the pixel-wise $L_{1}$ loss. The loss function $L_{1}$ is minimized using the Adam optimization algorithm [42], which is designed for first-order gradient-based optimization of stochastic objective functions. In the proposed framework, there are three branch learning processes. In each branch, the low-frequency image is denoted by $R_{LL}^{n},\;$ and the high-frequency image components are denoted by $R_{LH}^{n},\;R_{HL}^{n},$ and $R_{HH}^{n}$. The corresponding moiré-free low-frequency image is denoted by $G_{LL}^{n},\;$ and the moiré-free high-frequency image is denoted by $G_{LH}^{n},\;G_{HL}^{n},$ and $G_{HH}^{n}$. The loss of all branch learning is expressed as follows:(4)$$L_{branch}={L_{1}(R}_{LL}^{n},\;G_{LL}^{n})+\sum_{n=1}^{3}\left(L_{1}\left(R_{LH}^{n},\;G_{LH}^{n}\right)+L_{1}\left(R_{HL}^{n},\;G_{HL}^{n}\right)+L_{1}\left(R_{HH}^{n},\;G_{HH}^{n}\right)\right).\;$$

For the demoiréing output image $\hat{O}$ processed by three branch learning processes, we apply $L_{1}$ loss, feature-based perceptual loss $L_{p}$ [43], and Fourier space loss $L_{ft}$ [20]. The demoiréing loss function is denoted as follows:(5)$$L_{D}=L_{1}\left(\hat{O},\;O\right)+L_{p}\left(\hat{O},\;O\right)+L_{ft}\left(\hat{O},\;O\right),\;.\;$$
where O is the ground-truth image.

Then, the total loss of the proposed framework is formulated as follows:(6)$$L_{total}=L_{branch}+L_{D}.\;$$

## 4. Experimental Result

To quantitatively and qualitatively evaluate the performance of the proposed multibranch wavelet-based single image demoiréing method, the five well-known methods, including the latest learning-based moiré removal methods such as DMCNN [6], WDNet [15], MBCNN [14], EAFM [16], and UHDM [9], are compared. Finally, to investigate the effectiveness of different components in the proposed framework, we conducted an ablation study. 

### 4.1. Experimental Details and Datasets

The proposed method was implemented in Python programming language with PyTorch 1.7.1 [44] on a computer equipped Intel i7-10700F CPU, 2.9 GHz, 32 GB memory (Intel, Santa Clara, CA, USA), and NVIDIA GeForce RTX 3090 GPU (Nvidia, Santa Clara, CA, USA). The input image is 256 × 256 with a batch size of 4. The learning rate in our experiments is initially set to 0.0002 and decayed using cyclic cosine annealing [45], and the model is optimized using Adam method. To train the proposed MBWDN, the TIP dataset with a large diversity of moiré images [6] was employed. TIP includes 135,000 moiré images acquired from screens and captured under various imaging conditions, where each image pair contains one moiré image and its moiré-free ground-truth. We picked 90% images from the dataset for training, and the remaining 10% were used for testing.

### 4.2. Quantitative Evaluation

To quantitatively evaluate the performance of multibranch wavelet-based single image demoiréing method, the five moiré removal networks, including DMCNN, WDNet, MBCNN, EAFM, and UHDM, were used for comparisons, where all methods are also deep learning-based approaches. The three well-known metrics, peak signal-to-noise ratio (PSNR) [46], structural similarity index (SSIM) [47], and learning perceptual image block similarity (LPIPS) [48], were used for quality assessment of moiré-removed images. Quantitative comparison results are shown in Table 1. As shown in Table 1, the proposed method significantly outperforms the DMCNN, WDNet, and EAFM and outperforms the MBCNN and UHDM methods. Our method outperforms all methods, achieving a PSNR improvement of 1.11dB when compared to the state-of-the-art method, UHDM. Furthermore, the proposed method produces structures similar to moiré-free images with an SSIM value of 0.951. Among all compared methods, the number of parameters of the proposed network is slightly higher than that of DMCNN and WDNet; however, DMCNN and WDNet fail to effectively eliminate moiré patterns, resulting in the much lower PSNR values.

Most existing methods usually only address moiré effect in the frequency or spatial domain. However, the variety and complexity of moiré patterns makes it extremely challenging to completely remove them from a domain while preserving the original texture. Moiré patterns span low and high frequencies in the frequency domain and are intertwined with image texture in the spatial domain. In our multibranch architecture, the wavelet decomposes an image into different scales of details and approximations, allowing for multi-resolution analysis of the image. In general, the wavelet does not result in the loss of spatial domain information, so our proposed method can simultaneously remove moiré patterns both in the frequency domain and the spatial domain. Furthermore, we train two sub-networks to handle high and low frequencies, allowing each sub-network to focus on its designated task to obtain better demoiréing results.

### 4.3. Qualitative Evaluation

Figure 4 and Figure 5 illustrate the qualitative comparison of the different methods. We can observe that the input image is heavily contaminated by moiré patterns, showing obvious color changes and curve- or stripe-shaped patterns superimposed on the clear image. DMCNN does not effectively remove moiré patterns and has color distortion issues. UHDM shows better results in moiré removal and color restoration; however, in some cases, certain patterns still cannot be eliminated. In contrast, our proposed method is proven to be more effective in removing moiré patterns. Moreover, our proposed network performs well in recovering high-frequency details and edges because we separate high-frequency images in DMRN for training. Furthermore, in order to reveal the effectiveness of the proposed wavelet decomposition-guided image demoiréing framework in benefiting image detail restoration, we highlight some regions (highlighted by the red boxes) in the restored image, as shown in Figure 5. Based on the corresponding highlighted areas, the proposed method is more effective in reconstructing areas with rich texture, retaining more details, and providing better color recovery and better contrast than the compared methods. Furthermore, in order to demonstrate the practicality and feasibility of our proposed method, we used our proposed method on real moiré images produced by capturing LCD screens using mobile phone cameras, as shown in Figure 6. The recaptured screen images contain various types of moiré patterns. Although these moiré images are not included in the datasets used for evaluation, our proposed method demonstrates its capability to effectively eliminate various types of moiré patterns. The proposed method demonstrates its credibility by effectively enhancing the visibility of the recaptured screen image by removing moiré patterns.

### 4.4. Ablation Experiment

To investigate the effectiveness of the different components in the proposed framework, an ablation study is performed and described as follows. First, to evaluate the effectiveness of the multibranch structure, we remove two branches from the original structure. Second, to evaluate the effectiveness of the two networks used in the multibranch structure, we replace DMRN with MRN, and both sub-images are processed using MRN. Third, to evaluate the effectiveness of the loss function, we removed the loss of each branch $L_{D}$, and only $L_{1}$, $L_{p},\;$ and $L_{ft}$ are used between the final demoiréd image and the ground-truth image. Fourth, we removed $L_{ft}$ to verify its effectiveness. As shown in Table 2, replacing or removing these key components of the proposed results in poor image restoration quality.

## 5. Conclusions

This paper proposes a new multibranch wavelet-based image demoiréing network (MBWDN) embedded into wavelet image decomposition-guided framework for single image moiré pattern removal. Our method first decomposes an input moiré image into the low-frequency and high-frequency components. The low-frequency component is fed into our moiré removal network (MRN) with multiscale CNN, an attention module, and residual blocks to obtain the moiré-removed component while extracting the moiré-relevant structural features for preserving the structure and color information of the image. The high-frequency component is enhanced using residual dense blocks while preserving the original image details. Through multibranch learning, the moiré patterns of the decomposed images at all levels can be effectively removed while retaining the structure and color of the original image. The presented experimental results show that the proposed method achieves better moiré removal performance and lower (or comparable) computational complexity with the state-of-the-art single-image demoiréing algorithms.

The images in the test dataset contain moiré patterns throughout the image, but typically only a portion of the image is affected by moiré artifacts. Occasionally, the network cannot accurately identify moiré patterns. Furthermore, if an object or background in an image has a texture pattern similar to moiré artifacts, it is difficult for our method to distinguish whether it is a moiré pattern or not. Therefore, further improvements to the existing architecture are still needed. In the future, we will focus on improving the network architecture to better recognize for the above cases.

## Figures and Tables

**Figure 1 sensors-24-02762-f001:**
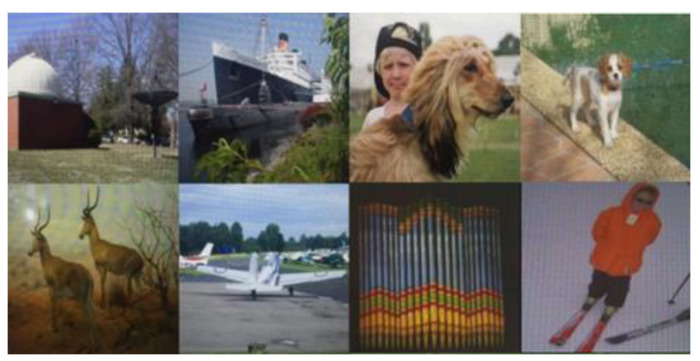
Moiré textures of different scales, frequencies, and colors.

**Figure 2 sensors-24-02762-f002:**
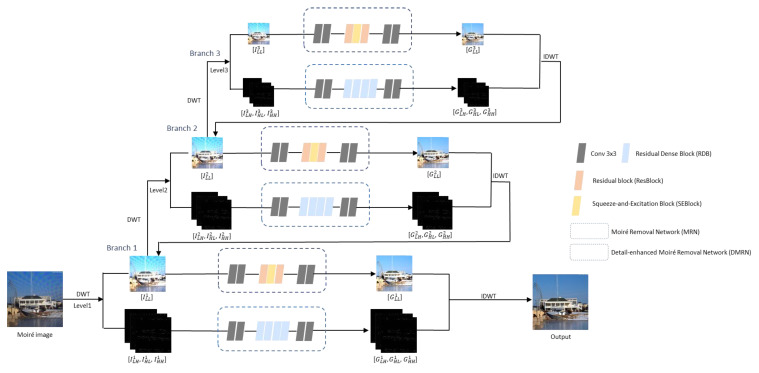
Illustration of the network architecture of our proposed method.

**Figure 3 sensors-24-02762-f003:**
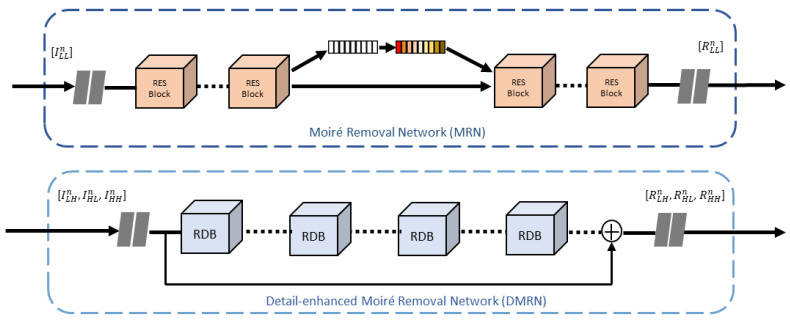
Illustration of the network architecture of the proposed MRN and DMRN.

**Figure 4 sensors-24-02762-f004:**
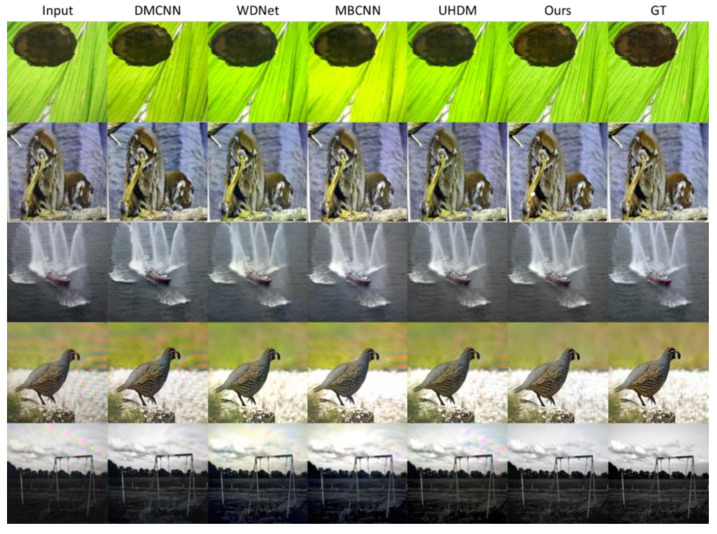
Comparisons of visual quality among DMCNN, WDNet, MBCNN, and UHDM.

**Figure 5 sensors-24-02762-f005:**
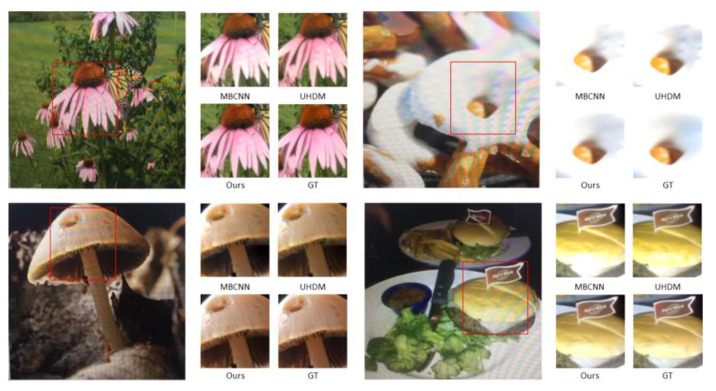
Comparisons of visual quality evaluation of complex patterns areas among DMCNN, WDNet, MBCNN, and UHDM.

**Figure 6 sensors-24-02762-f006:**
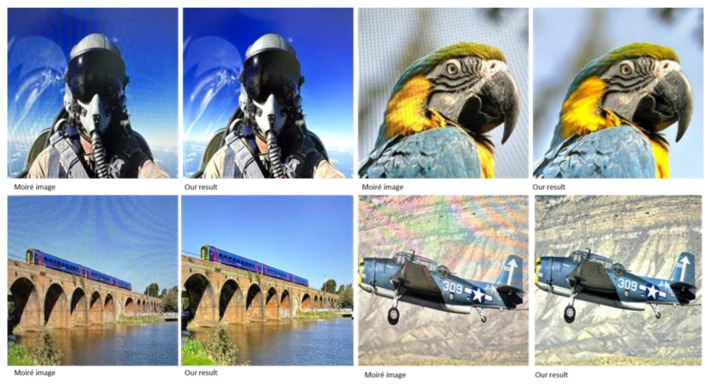
Real moiré images and the corresponding outputs of our proposed method.

**Table 1 sensors-24-02762-t001:** Comparisons of quantitative results evaluated in terms of average PSNR (dB), SSIM, and LPIPS. (↑) indicates greater is preferable, while (↓) indicates lesser is preferable.

Dataset	Metrics	Input	DMCNN	WDNet	MBCNN	EAFN	UHDM	Our Method
TIP dataset	PSNR ↑	20.30	26.77	28.08	30.03	29.70	30.11	31.22
	SSIM ↑	0.738	0.871	0.904	0.893	0.893	0.920	0.951
	LPIPS ↓	0.191	0.142	0.164	0.096	-	0.082	0.077
LCDMoiré	PSNR ↑	10.44	35.48	29.66	44.04	-	45.34	45.82
	SSIM ↑	0.572	0.979	0.967	0.995	-	0.997	0.998
	Params (M)	-	1.426	3.360	14.192	38.000	10.623	8.071

**Table 2 sensors-24-02762-t002:** Results of the ablation study.

	Only One Branch	Only MRN Network	Only $L_{D}$ Loss	w/o FFT Loss	Full Method
PSNR	29.96	29.85	30.74	30.92	31.22
SSIM	0.936	0.935	0.943	0.947	0.951

## Data Availability

Data are contained within the article.

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
