# Peer review of "Multibranch Wavelet-Based Network for Image Demoiréing"

_sensors, 2024, doi:10.3390/s24092762_

Round 1

Reviewer 1 Report

Comments and Suggestions for Authors

1. The use of wavelet-based image restoration has been widely proposed, with some closely related works overlooked by certain authors, such as "Toward Universal Stripe Removal Via Wavelet-based Deep Convolutional Neural Network." Additionally, while embedding wavelet transforms into deep networks for moiré pattern removal has been done, the authors did not provide a clear explanation of the benefits and advantages of this approach. It is suggested to enhance the motivation analysis of using wavelet transforms to address moiré patterns from both experimental and theoretical perspectives.

2. The experimental section lacks sufficient persuasiveness. The authors only train and test on a simulated dataset, which does not demonstrate the practicality of the method. It is suggested to include experimental results comparing real-world moiré suppression for increased credibility. Additionally, there is insufficient ablation analysis and discussion. For instance, whether the claimed benefits of wavelet decomposition and moiré suppression in different frequency bands hold true as stated in the paper lacks support from intermediate experimental results.

3. The resolution of the images in the paper is too low, to the extent that in many images, moiré patterns are barely visible. The visualization of the contrast results is very poor, making it difficult to distinguish between good and bad outcomes. For example, in Figure 5, no details can be seen at all, and the images are blurry and low-resolution, far exceeding the moiré pattern issue that the paper aims to address. We strongly recommend that the authors recreate the images with higher DPI to improve visual clarity.

Comments on the Quality of English Language

can be improved 

Author Response

  1. The use of wavelet-based image restoration has been widely proposed, with some closely related works overlooked by certain authors, such as "Toward Universal Stripe Removal Via Wavelet-based Deep Convolutional Neural Network." Additionally, while embedding wavelet transforms into deep networks for moiré pattern removal has been done, the authors did not provide a clear explanation of the benefits and advantages of this approach. It is suggested to enhance the motivation analysis of using wavelet transforms to address moiré patterns from both experimental and theoretical perspectives.

ANS:

Thank you for your comments. Wavelet transform has several advantages including time-frequency localization, multi-resolution analysis and noise reduction. For time-frequency localization, wavelets can be used to identify specific features in a signal, such as transient events or harmonics, that may not be visible with other time-frequency localization methods. Moiré patterns are characterized by higher harmonics and transient events, and wavelet transform is useful for identifying these specific features. The wavelet transform provides a multi-resolution representation of the signal, which means that the Moiré signal can be analyzed at different resolution levels so that we can use different strategies to process it. Moreover, wavelet transform can separate signals and noise in different frequency bands, making it easier to suppress noise in Moiré patterns. The proposed multi-branch network combined with wavelet transform is able to extract more meaningful and sparse moiré pattern representations.   

The above discussion has been incorporated into the revised manuscript.

  1. The experimental section lacks sufficient persuasiveness. The authors only train and test on a simulated dataset, which does not demonstrate the practicality of the method. It is suggested to include experimental results comparing real-world moiré suppression for increased credibility. Additionally, there is insufficient ablation analysis and discussion. For instance, whether the claimed benefits of wavelet decomposition and moiré suppression in different frequency bands hold true as stated in the paper lacks support from intermediate experimental results.

ANS:

Thank you for your comments. To demonstrate the practicality and feasibility of our proposed method, we conducted our proposed method on real moiré images produced by capturing LCD screens using mobile phone cameras. The restoration results are shown in Fig. 6 of the revised manuscript. It is evident that the proposed method effectively eliminates moiré patterns from moiré images, thereby enhancing visibility and demonstrating its credibility. The following paragraph is added in the revised manuscript.

Furthermore, in order to demonstrate the practicality and feasibility of our proposed method, we conducted our proposed method on real moiré images produced by capturing LCD screens using mobile phone cameras. The recaptured screen images contain various types of moiré patterns. Although these moiré images are not included in the datasets used for evaluation, our proposed method demonstrates its capability to effectively eliminate various types of moiré patterns. The proposed method demonstrates its credibility by effectively enhancing the visibility of the recaptured screen image through removing moiré patterns.

In the ablation study, we designed two scenarios to investigates the performance of the proposed multi-branch wavelet-based image demoiréing network. One is to replace the multibranch architecture with a single branch to understand the contribution of wavelet multi-resolution representation. The other is to replace DMRN with MRN in a multibranch architecture to demonstrate that the design of these two components is effective for moiré removal of low- and high-frequency patterns. Experimental results shows that single branch architecture fail to effectively remove moiré patterns. Furthermore, MRN alone does not preserve the details of the restored images well. The results in both cases significantly degrade the demoiréing performance in terms of PSNR and SSIM.

  1. The resolution of the images in the paper is too low, to the extent that in many images, moiré patterns are barely visible. The visualization of the contrast results is very poor, making it difficult to distinguish between good and bad outcomes. For example, in Figure 5, no details can be seen at all, and the images are blurry and low-resolution, far exceeding the moiré pattern issue that the paper aims to address. We strongly recommend that the authors recreate the images with higher DPI to improve visual clarity.

ANS:

Thank you for your comments. We recreated the image in Figure 5 following the reviewer's comments to enhance image resolution for better visual clarity.

Reviewer 2 Report

Comments and Suggestions for Authors

I have the following concerns.

1. Various metrics are used to assess image quality (IQA). In addition to SSIM, PSNR, DSSIM, MSE, and FSIM are also known. Therefore, Table 1 should be supplemented with new comparisons.

2. It is necessary to show the limitations of your approach to removing moiré on images.

3. Justify which discrete wavelet transformation was used by Haars, Doubechies, Mallats.

4. Justify that DWT will not be inferior to DCWT in terms of efficiency.

5. Supplement References with articles for 2022-2024 to confirm the relevance of research on eliminating the effect of moire on image quality.

Comments on the Quality of English Language

 Minor editing of English language required

Author Response

  1. Various metrics are used to assess image quality (IQA). In addition to SSIM, PSNR, DSSIM, MSE, and FSIM are also known. Therefore, Table 1 should be supplemented with new comparisons.

ANS:

Thank you for your comments. We include new evaluations based on reviewer comments for comparison. The Learning Perceptual Image Block Similarity (LPIPS) metric is designed to evaluate the perceptual similarity between two images. This measurement has been shown to match human perception well, and low LPIPS values ​​indicate that image patches are perceptually similar. From Table 1 of the revised manuscript, the proposed method still has superior performance compared with other existing methods.

  1. It is necessary to show the limitations of your approach to removing moiré on images.

ANS:

Thank you for your comments. We have added limitations and future plans of the proposed method in the conclusion section. The following discussion has been incorporated into the revised manuscript.

The images in the test dataset contain moiré patterns throughout the image, but typically only a portion of the image is affected by moiré artifacts. Sometimes the network cannot accurately identify moiré patterns. Furthermore, if an object or background in an image has a texture pattern similar to moiré artifacts, it is difficult for our method to distinguish whether it is a moiré pattern or not. Therefore, further improvements to the existing architecture are still needed. In the future, we will focus on improving the network architecture to better recognize for the above cases.

  1. Justify which discrete wavelet transformation was used by Haars, Doubechies, Mallat.

ANS:

Thank you for your comments. The main difference between these wavelet functions is the smoothness of the wavelet. Haar wavelets are a special case of Daubechies wavelets, Daubechies average over more pixels and are smoother than Haar wavelets. Mallat is generally smoother because it is based on successive iterations of a low-pass filter. Haar wavelets are often used to detect sudden changes because their basic function exhibits clear discontinuities, allowing them to capture sudden changes or jumps in the signal. This property makes Haar wavelets perform well in various signal processing tasks, such as edge detection. In comparison, other wavelets such as Daubechies etc. are commonly used for smoothing images and texture analysis. This is because their basis functions are smoother and can capture the local structure and texture characteristics of the signal without producing obvious discontinuities. From our perspective, moiré patterns often resemble sudden changes because moiré patterns are caused by interference between two or more grid structures, resulting in noticeable gaps or stripes in the image. These stripes are usually caused by phase differences or overlap, rather than natural variations in texture. In summary, the Haar wavelet would be a better choice for finding the location of edges. Therefore, we choice the Haar transform in our multibranch architecture to solve moiré patterns problem.

  1. Justify that DWT will not be inferior to DCWT in terms of efficiency.

ANS:

Thank you for your comments. Compared with DCWT, DWT may be more effective in handling non-stationary signals and sudden changes. Since DWT can capture local features of signals at different scales, it may be more suitable for processing mutated signals. Furthermore, DWT generally has lower computational complexity because it only requires the calculation of a limited number of wavelet coefficients, whereas DCWT requires the calculation of a large number of cosine coefficients. This makes DWT faster and more efficient in some cases.

  1. Supplement References with articles for 2022-2024 to confirm the relevance of research on eliminating the effect of moire on image quality.

ANS:

Thank you for your comments. In the revised manuscript, we have properly cited the suggested references (articles for 2022-2024) with related discussions. There are three papers on the similar topic which are highly relevant shown in the following. We have carefully surveyed these papers and made the discussion in Section 2.2.

11.Yang, C.; Yang Z.; Ke K.; Chen T.; Grzegorzek M.; See J. Doing More With Moiré Pattern Detection in Digital Photos. IEEE Transactions on Image Processing, 2023, pp.694-708.

  1. Niu, Y.; Lin Z.; Liu, W.; Guo, W. Progressive Moire Removal and Texture Complementation for Image Demoireing. IEEE Transactions on Circuits and Systems for Video Technology, 2023.
  2. Nguyen, D. H.; Lee, S.; Lee, C. Multiscale Coarse-to-Fine Guided Screenshot Demoiréing. IEEE Signal Processing Letters, 2023

Reviewer 3 Report

Comments and Suggestions for Authors

The authors proposed a method for image Demoireing using CNNs on the Wavelet coefficients. The explanation of the method is quite clear ,and the proposed network seems reasonable. but in my opinion there are some concerns on evaluation of the results and supporting evidences.

1. The method is only compared on one dataset TIP dataset. However, many other methods like UHDM are evaluated on at least 2 other datasets such as LCDMoire as well. Is there any specific reason not to test your method on those datasets?

2. It would be interesting to explain more on why the method is better compared to other method, at least intuitively. I did not find any disscusion on this in the paper. 

Comments on the Quality of English Language

3. I think english writing of the paper should be improved, for example the sentence in line 200 "for the high-frequency and low-200 frequency sub-images of level 𝑛 will be processed by the branch 𝑛 learning;"  the word learning could be removed. I think the text from line 200 to 205 should be rewritten.

Author Response

  1. The method is only compared on one dataset TIP dataset. However, many other methods like UHDM are evaluated on at least 2 other datasets such as LCDMoire as well. Is there any specific reason not to test your method on those datasets?

ANS:

Thank you for your comments. We follow the reviewer’s comment to add another dataset for evaluation. We supplement the evaluation with the LCDMoire dataset. In the revised manuscript, Table 1 show that the proposed method still has the best demoiréing compared to other existing methods.

  1. It would be interesting to explain more on why the method is better compared to other method, at least intuitively. I did not find any discussion on this in the paper.

ANS:

Thank you for your comments. Most existing methods usually only address moiré effect in the frequency or spatial domain. However, the variety and complexity of moiré patterns makes it extremely challenging to completely remove them from a domain while preserving the original texture. Moiré patterns span low and high frequencies in the frequency domain and are intertwined with image texture in the spatial domain. In our multibranch architecture, the wavelet decomposes an image into different scales of details and approximations, allowing for multi-resolution analysis of the image. In general, the wavelet does not result in the loss of spatial domain information so our proposed method can simultaneously remove moiré patterns both in the frequency domain and the spatial domain. Furthermore, we train two sub-networks to handle high and low frequencies, allowing each sub-network to focus on its designated task to obtain better demoiréing results.

The above discussion has been incorporated into the revised manuscript.

  1. I think English writing of the paper should be improved, for example the sentence in line 200 "for the high-frequency and low-200 frequency sub-images of level ? will be processed by the branch ? learning;" the word learning could be removed. I think the text from line 200 to 205 should be rewritten.

ANS:

Thank you for your comments. We have revised the text from line 200 to 205 in the revised manuscript and shown below.  

For example, at the level n, both high-frequency and low-frequency sub-images are processed by the pipeline process, branch n. The output of branch n includes the low-frequency sub-image of level n-1, which is processed by branch n-1 together with the high-frequency sub-image of level n-1. This looping process continues on all branches until the output size matches the original input, thereby enhancing the network's ability to capture complex features and patterns. The multibranch learning process can be represented as:

Round 2

Reviewer 2 Report

Comments and Suggestions for Authors

I am almost satisfied with the answers and additions, except for the first one.

Comments on the Quality of English Language

 Minor editing of English language required

Author Response

Reviewer2:

I am almost satisfied with the answers and additions, except for the first one.

Minor editing of English language required.

Ans:

Thanks for your comments. Thank you for providing insightful suggestions for the improvement of our work. We have read the comments carefully and tried to follow the suggestions as closely as possible. Grammatical and writing style errors in the original version have been corrected by a native English speaker. The changes are highlighted in gray in the revised manuscript.

Reviewer 3 Report

Comments and Suggestions for Authors

All the comments are addressed in the new version and the manuscript improved noticeably in terms of presentation of the method and the results.

Author Response

Reviewer3:

All the comments are addressed in the new version and the manuscript improved noticeably in terms of presentation of the method and the results.

Ans:

Thanks for your comments. We are pleased that our revised manuscript has addressed your comments We would like to thank you for your time and efforts reading our manuscript, providing insightful suggestions for the improvement of our work.